# Effects of bile salt-stimulated lipase on blood cells and associations with disease activity in human inflammatory joint disorders

Susanne Lindquist[1,2],* , Yuhang Wang[1,¤a], Eva-Lotta Andersson[1,2], Shizuko Tsuji Grebe[1,¤b], Gerd-Marie Alenius[3], Solbritt Rantapää-Dahlqvist[3], Lennart Lundberg[2], Olle Hernell[1,2]

1 Department of Clinical Sciences, Pediatrics, Umeå University, Umeå, Sweden, 2 Lipum AB, Umeå, Sweden, 3 Department of Public Health and Clinical Medicine, Rheumatology, Umeå University, Umeå, Sweden

☯ These authors contributed equally to this work.
¤a Current address: Department of Pathology, University of Iowa, Iowa, IA, United States of America
¤b Current address: Department of Pediatrics, Nara Medical University, Kashihara, Nara, Japan
* susanne.lindquist@lipum.se

## Abstract

The bile salt-stimulated lipase (BSSL) was originally recognized as a lipolytic enzyme expressed by the exocrine pancreas and in some species, notably humans, the lactating mammary gland, being secreted into the duodenum and with the mother's milk, respectively. However, BSSL is also present in the blood and has been assigned additional functions, even beyond the gastrointestinal tract. Conventional BSSL knockout mice are protected from developing disease in animal models of arthritis, and antibodies directed towards BSSL prevent or mitigate disease in similar models. The aim of this study was to investigate the role of BSSL as a newly discovered player in inflammation and specifically in inflammatory joint disorders. As part of mechanism of action, we here show that BSSL is secreted by neutrophils, interacts with monocytes and stimulates their migration *in vitro*. An anti-BSSL antibody that blocks the human BSSL-monocyte interaction was shown to simultaneously prevent the signaling pathway by which BSSL induce cell migration. Moreover, in this cohort study we show that BSSL levels are significantly higher in blood samples from patients with rheumatoid arthritis and psoriatic arthritis compared to healthy controls. The BSSL levels in patients' blood also correlated with disease activity scores and established inflammatory markers. Hence, although the mode of action is not yet fully clarified, we conclude that BSSL could be considered a proinflammatory component in the innate immune system and thus a possible novel target for treatment of chronic inflammation.

## Introduction

Bile salt-stimulated lipase (BSSL), also termed carboxyl ester lipase (CEL), bile salt-dependent lipase (BSDL), bile salt-activated lipase (BAL) or carboxyl ester hydrolase (CEH), is a protein initially recognized as a lipolytic enzyme secreted from exocrine pancreas and in some species,

**Data Availability Statement:** All relevant data are within the paper and its Supporting Information files.

**Funding:** This study was sponsored by Lipum AB, including salary to SL and ELA. Financial support was also received from Västerbottens läns landsting (SL, OH), Umeå Biotech Incubator (SL, LL, OH) and Oskarfonden (SL). YW was sponsored by the China Scholarship Council. The funders had no role in study design, data collection and analysis, decision to publish, or preparation of the manuscript.

**Competing interests:** SL, LL and OH are scientific co-founders of Lipum, a biopharmaceutical company focused on novel treatment of chronic inflammatory diseases. They hold shares and have rights to intellectual property of the company (WO/ 2010/117325, New methods for treatment of inflammatory diseases; WO/2021/010888, Novel BSSL antibodies). SL is employed CSO of Lipum. OH serves on Lipum's board of directors, LL and SRD serves on Lipum's scientific advisory board. They report remuneration from Lipum. ELA is employed and holds shares in Lipum. This does not alter our adherence to PLOS ONE policies on sharing data and materials. YW, STG and GMA has no competing interest to declare.

notably humans, also from the lactating mammary gland to be a constituent of the milk [1, 2]. The milk BSSL is a multifunctional protein that contributes to efficient utilization of milk fat in breast fed infants [3] but it has also been linked to the protective effect of mother's milk against viral infections [4–6]. The presence of BSSL is not confined to the gastrointestinal tract, it is also found in peripheral blood [7]. The concentration in serum from healthy humans is $1.5 \pm 0.5$ µg/L but it is elevated to levels as high as 7 µg/L in some pathologic conditions, such as acute pancreatitis [8, 9]. The source of circulating BSSL is not fully understood. The protein has been suggested to be expressed by macrophages [10], endothelial cells [11] and to be stored in platelets and released upon their activation to affect thrombus formation [12]. The BSSL in blood may also, at least in part, originate from protein secreted by the pancreas and reabsorbed by the intestine [13]. However, there are conflicting data and therefore it is not entirely clear whether and, if so, to what extent it may be absorbed [7, 14].

In addition to its roles in dietary fat digestion and protection against viral infections, BSSL has been discovered as an essential player in inflammatory arthritis [15]. Conventional BSSL knockout mice are significantly less prone to develop collagen-induced arthritis (CIA) compared to wildtype littermates, and antibodies directed towards BSSL has been shown to mitigate disease activity in two different species and experimental models of arthritis, i.e. CIA in mice and pristane-induced arthritis in rats [15]. We hypothesized that BSSL is involved in the inflammatory process not only in rodents, but also in patients with rheumatoid arthritis (RA) and psoriatic arthritis (PsA) and envision humanized anti-BSSL antibodies to be as a novel therapy to combat arthritis.

RA and PsA are two immune-mediated inflammatory diseases that primarily affect joints and cause pain, swelling and stiffness. Many features associated with RA and PsA overlap but there are notable differences, most obvious that PsA also involves psoriasis of the skin, which is a persistent chronic disease in itself. Moreover, while RA is characterized as a chronic destructive disease predominantly determined by the adaptive immune response, PsA is considered a mixed patterned autoimmune/autoinflammatory disease with features related to both adaptive and innate immune responses [16]. Although important progress has been made in recent years in unravelling genetic and environmental factors, the exact causes of RA and PsA remain elusive [17–19]. The goal for modern treatment in RA and PsA is to achive remission or at least low disease activity, which is strongly supported by the pharmacological development during the last 2 decades with different treatment options [20].

The aim of the present study was to explore interactions and biological effects of BSSL on human inflammatory cells, i.e. primary granulocytes, monocytes and lymphocytes. A second aim was to determine BSSL levels in blood samples from patients with RA and PsA in order to elucidate possible associations with disease activity in human inflammatory joint disorders.

## Material and methods

### Experimental in vitro studies

**Purification and biotin labelling of native human milk BSSL.** Human breast milk was collected from several donors at different time points during lactation and stored frozen at -20˚C. The BSSL protein was isolated essentially as previously described [21, 22]. The peak fractions containing BSSL lipase activity were collected and analyzed for integrity and purity by SDS-PAGE. For biotin labelling of native BSSL the EZ-Link Alkoxyamine-PEG4-Biotin Reagents (Thermo Fisher Scientific, Waltham, MA, USA) was used according to the supplier's instruction.

**Development of anti-human BSSL antibodies.** The monoclonal mouse anti-human BSSL antibody AS20 was developed by Agrisera AB (Vännäs, Sweden) on behalf of the Hernell

lab. at Umeå University. In brief, Balb/C mice were immunized intraperitoneally with BSSL protein isolated from human breast milk (see above). Mice were boosted twice with BSSL protein emulsified with Freund's incomplete adjuvant wherafter they were sacrificed. Spleen cells were harvested and fused with mouse myeloma SP2/0 cells using a standard protocol. A hybridoma cell line secreting antibodies against native BSSL protein was identified. This cell line was subcloned to establish a stable clone expressing mouse monoclonal, anti-human BSSL antibodies of IgG1 subclass. The clone was denoted AS20.

The polyclonal rabbit anti-human BSSL antibody Unti was also developed by Agrisera AB. Antibodies were raised in rabbits by immunization with a synthetic peptide corresponding to amino acid 328–341 in the mature human BSSL protein [23]. Monospecific anti-peptide 328–341 antibodies were purified from serum using a peptide affinity column obtained from Agrisera AB according to the suppliers' protocol.

## Isolation of human peripheral blood cells and fMLP stimulation

Peripheral blood from healthy volunteers were drawn in vacutainer EDTA tubes. Polymorphonuclear granulocytes were isolated using Polymorphprep™ (Axis-Shield, Oslo, Norway) and following the manufacturer's instruction. Granulocytes ($1x10^6$ cells) in 500 μl RPMI 1640 supplemented with L-glutamine and Penicillin/Streptomycin Solution (Thermo Fisher Scientific) were incubated at 37˚C in the presence of N-Formyl-Met-Leu-Phe peptide, between 0.2–5.0 μM (Sigma-Aldrich, St Louis, MO, USA). At different time points, cells were removed by centrifugation and BSSL concentration in the culture media was determined by ELISA.

Peripheral blood mononuclear cells (PBMCs) were isolated from healthy volunteers using Ficoll-Paque PLUS solution (Cytiva, Uppsala, Sweden) according to manufacturer's instruction. The PBMCs were suspended in RPMI 1640 supplemented with L-glutamine and Penicillin/Streptomycin Solution (Thermo Fisher Scientific). In cases where the PBMCs were intended for further sub-fractionation (monocyte isolation), the cells were suspended in PBS-EDTA. Monocytes were isolated from the PBMC fraction by negative selection using the monocyte isolation kit II (Miltenyi Biotech, Lund, Sweden) according to manufacturer's instruction. The monocytes were suspended in RPMI 1640 supplemented with L-glutamine and Penicillin/Streptomycin Solution (Thermo Fisher Scientific).

## Flow cytometry

Peripheral blood drawn in vacutainer tubes supplemented with sodium citrate as anticoagulant or freshly prepared buffy coat were used for flow cytometry. The samples (whole blood or buffy coat) were stained with fluorochrome-conjugated mouse anti-human monoclonal antibodies (CD14-BV421, CD15-PE-Cy7 and CD3 PE-Cy5) from BD Biosciences (Franklin Lakes, NJ, USA). Mouse anti-human BSSL mAb AS20 was conjugated to Alexa Fluor 647 (AS20-AF647) using the Alexa Fluor® Antibody Labeling Kit (Molecular Probes by Life Technologies, Thermo Fisher Scientific), according to the manufacturer´s instruction, and used to detect BSSL. Exogenous biotin-labelled BSSL (bio-BSSL) was detected using BD Horizon BB515 Streptavidin (BD Biosciences). To analyze antigens on the cell surface, the protocol for direct immunofluorescence staining of whole blood using a lyse/wash procedure was used, as previously described (www.bdbiosciences.com). To analyze intracellular markers, the cells were first permeabilized using BD FACS™ permeabilizing solution 2 (BD Biosciences) before staining was performed following the manufacturer's instruction. Flow cytometry was performed on a FACS LSR II (BD Biosciences) and data were analyzed using FlowJo software (BD Biosciences). Leukocyte populations were defined as CD14+ monocytes, CD15+ granulocytes and CD3+ T lymphocytes, respectively.

## Immunofluorecence staining and confocal imaging

Human peripheral leukocytes were isolated from healthy volunteers using the Polymorph-prep™ (Axis-Shield), according to manufacturer's instruction. The cells were applied in a drop of 10 μl onto SuperFrost Plus slides (Menzel-Gläser, Braunschweig, Germany) and allowed to settle for 1 h at room temperature (RT) in a humidified chamber. Immunofluorescence stainings were done essentially as previously described [24]. In brief, cells were fixed using 4% para-formaldehyde in 0.1 M phosphate buffer (pH 7.0) for 20 min at RT. To permeabilize the cells and block for unspecific binding, slides were incubated in 1% Triton X-100 in 0.1 M Tris-HCl pH 7.5, 1.5 M NaCl (TBS-T) supplemented with 10% normal horse serum (NHS) for 1 hour at RT. Primary antibodies, diluted in TBS-T + 10% NHS, were applied and incubated for 2 h at RT. Cells were then washed in TBS-T to remove the excessive primary antibodies followed by incubation with fluorescent secondary antibodies (Alexa fluor 488 goat-anti-mouse IgG or Alexa fluor 594 goat-ant-rabbit IgG; Molecular Probes, Eugene, Oregon, USA) for 1 hour at RT. 4′,6-diamidino-2-phenylindole (DAPI, Molecular Probes) was used for nuclear staining. For staining non-permeabilized cells, TBS-T was replaced by PBS in all steps. Images were taken by using a Zeiss LSM 710 confocal microscope.

## Binding of exogenous BSSL to human white blood cells and anti-BSSL antibody displacement assay

Binding of exogenous BSSL to human white blood cells was analyzed by incubating bio-BSSL (1 μg) with 100 μl freshly prepared buffy coat (resulting in approx. 0.1 μM BSSL) for 30 min at 4˚C. The cells were then washed and stained with BD Horizon BB515 Streptavidin (BD Biosciences) to detect and quantify binding of bio-BSSL to different cellular subsets, using flow cytometry as described above. To investigate the capacity of mAb AS20 or an isotype control antibody to block (displace) binding of bio-BSSL to human white blood cells, the bio-BSSL was preincubated with mAb AS20, an isotype control antibody or vehicle alone for 30 min at 4˚C before added to the buffy coat. The capacity of mAb AS20 or the isotype control to block (displace) binding of bio-BSSL to cellular subsets was quantified as a reduction in BB515 MFI compared to the vehicle (negative) control.

## Transwell migration assay

*In vitro* migration assays were performed in 24-well plates with polycarbonate Transwell inserts (diameter 6.5 mm; pore size 5.0 μm) (Corning Costar, Cambridge, MA). The lower chambers were filled with 600 μl of RPMI 1640 medium, supplemented with L-glutamine, penicillin/streptomycin solution (Thermo Fisher Scientific), and various test substances including BSSL. Freshly isolated PMBCs ($1 \times 10^6$ cells in 100 μl) were placed in the upper chamber of the Transwell device. In some of the experiments, the BSSL protein was pre-incubated with antibodies or with the lipase- and cholesterol esterase inhibitor WAY-121,898 [25] for 30 min at 37˚C, before added to the Transwell plate. Following an incubation period of 4 hours at 37˚C in 5% $CO_2$, the cells that had migrated through the semipermeable membrane were recovered and analyzed by flow cytometry.

## Patients and controls

The study included 15 patients diagnosed with RA (ACR criteria) [26] before initiating and during treatment with the TNFa inhibitor infliximab (Remicade®) (Presented in S1 Table). The patients received infliximab as intravenous injections (3 mg/kg) at baseline (day 1) and thereafter at 2 weeks, 6 weeks, 14 weeks, 22 weeks and 30 weeks. Blood (EDTA plasma) was

sampled at three time points, at baseline (before first dose), 14 weeks and either 22 or 30 weeks after the first dose. Clinical examination was assessed with the 28-joint count of tender and swollen joints, erythrocyte sedimentation rate (ESR) and with a global health visual analog scale at every visit. Disease activity score (DAS28) was calculated according to Prevoo [27] at baseline and at every visit when the 15 patients received their treatment. Data on C-reactive protein (CRP) levels, leukocyte- and neutrophil counts were only available from a subset of 8 patients at a total of 24 visits (at baseline, at 14 weeks, and at either 22 or 30 weeks).

The study also included 43 patients diagnosed with PsA and 28 healthy controls, matched for age and gender (S2 Table). High sensitive CRP (hs-CRP), S-calprotectin (S100A8/A9), cytokines and chemokines were analyzed in both patient and control groups, whilst measurements of ESR, hemoglobin (Hb), leukocyte- and thrombocyte counts, s-urate and s-creatinine were only performed for the PsA patients [28]. All patients were examined clinically for inflammatory joint manifestations and skin involvement. The number of tender and swollen joints, with a duration of more than 6 weeks, was assessed using 66-joint count. Mono-/oligoarthritic disease pattern was defined when four or less tender or swollen joints were present at the time of the examination, and polyarthritic disease pattern was diagnosed when more than four tender and swollen joints were present.

The study was approved by the Regional Ethics Committee of Umeå University under the license numbers Dnr 2016-130-32M, Admendment to Dnr 04-064M and Um Dnr 96–229. All participants were 18 years or older and gave their written informed consent. The patients were recruited to the study and the samples were collected in 2016 (RA) and from 1997 to 1999 (PsA), respectively. Only the study physicians responsible for clinical data aquisiton (SRD and GMA) had access to information that could identify individual participants. The analysis of the samples were performed without any knowledge of the individual participants during or after data collection. Blood samples were stored at -80˚C until analysis.

## Laboratory measurements

The BSSL concentration in culture media, plasma and serum was analyzed using an in-house sandwich enzyme-linked immunosorbent assay (ELISA), essentially as previously described [7]. In this study, the monoclonal mouse anti-human BSSL antibody AS20 (see above) was used as capture antibody and a biotinylated polyclonal rabbit anti-human BSSL antibody as detection antibody. BSSL´s enzymatic activity was determined as the capacity to hydrolyze radiolabeled triglyceride or cholesterol ester substrates as previously described [21, 22]. The cytokines in RA plasma were measured using Human IL-1β/IL-1F2, Human IL-6 and Human TNF-α DuoSet ELISA kits from R&D Systems (Minneapolis, MN, USA). Assays were performed according to the manufacturer's protocol and analyzed with a Multiskan GO and ScanIt Software 4.1 for Multiplate Readers (Thermo Fisher Scientific). In PsA patients and healthy controls, hs-CRP and calprotectin (S100A8/A9) were measured in serum whereas cytokines and chemokines were measured in plasma as previously described [28]. ESR, CRP, Hb, leukocyte-, neutrophil-, and thrombocyte counts, s-urea and s-creatinine were measured using routine methods at the Department of Clinical Chemistry, University Hospital of Umeå, Sweden.

## Statistical analyses and graphs

Spearman rank-order equation was used to analyze for correlations. Differences between two groups were tested using Mann-Whitney U tests and for multiple groups, the Kruskal-Wallis exact test. In case of an overall significant difference, pairwise Mann-Whitney U tests were subsequently performed. Wilcoxon matched-pairs signed rank tests were used to compare measurements taken from the same subject at multiple time points, i.e. before and during

treatment of RA patients with infliximab. Statistical analyses and graphs were performed using GraphPad Prism 9 (GraphPad Software, La Jolla, CA, USA) except for the non-parametric local regression (LOESS) curves with 95% confidence intervals that were created using R statistical software (v4.2.2; R Core Team 2021). Mean values are reported ± SEM unless otherwise stated. $P$-values $< 0.05$ were considered statistically significant.

## Results

### BSSL in human peripheral white blood cells

The previously reported presence of BSSL in human plasma was verified in a few healthy volunteers using ELISA (1.0 ± 0.2 µg/L; n = 3). Possible interactions between BSSL and peripheral white blood cells were then investigated using flow cytometry. Fluorescently labelled monoclonal anti-human BSSL antibodies (mAb AS20) detected BSSL on the surface of non-permeabilized CD15+ granulocytes in samples of whole blood from healthy subjects (Fig 1A and 1C). The mean fluorescence intensity (MFI) increased significantly in permeabilized cells, indicating intracellular location, mainly in the CD15+ granulocyte population but also some increase in CD14+ monocyte population (Fig 1B and 1C). There was no significant interaction detected between BSSL and CD3+ T lymphocytes, neither with intact, nor permeabilized cells (Fig 1A–1C).

Immunofluorescent staining of peripheral white blood cells with anti-BSSL antibodies followed by confocal imaging verified the presence, and especially the intracellular location of BSSL in these cells (Fig 2A–2H). The strongest signal was detected in CD15+ granulocytes, seemingly localized to membrane bound and intracellular granules (Fig 2I). Which type of granule is not yet clear, but isolated granulocytes stimulated with the chemotactic peptide N-formyl-Met-Leu-Phe (fMLP) *in vitro* were shown to respond by secreting BSSL into the culture medium (Fig 2J and 2K).

The interaction of BSSL with white blood cells was further investigated by incubating exogenous, biotin-labelled BSSL (bio-BSSL) with human buffy coat, freshly prepared from healthy donors. Binding of bio-BSSL to different subsets of white blood cells was analyzed using fluorescently labelled streptavidin in flow cytometry. The exogenous bio-BSSL bound primarily to the surface of CD14+ monocytes and to a lesser extent to CD15+ granulocytes (Fig 3A). No significant binding was detected to CD3+ T lymphocytes. The interaction between exogenously added bio-BSSL and CD14+ monocytes was significantly blocked by preincubating bio-BSSL

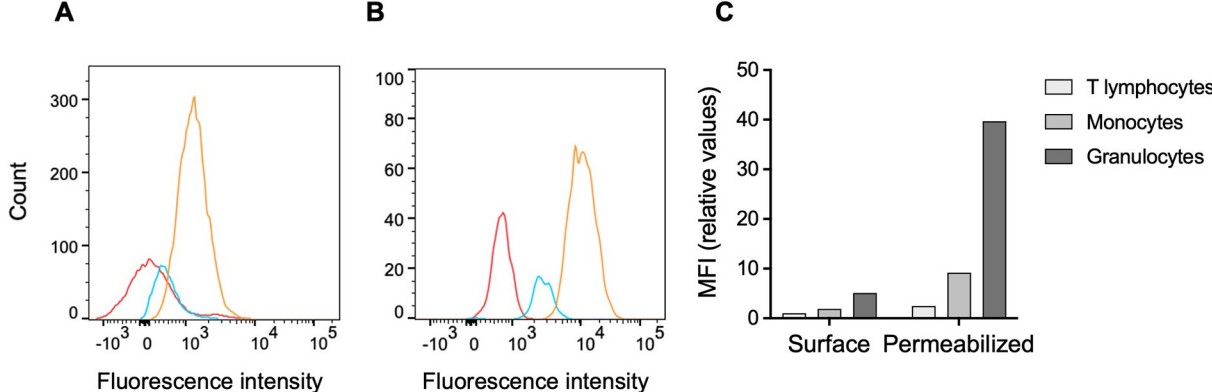

**Fig 1. BSSL in human peripheral blood cells.** Whole blood was incubated with fluorescently labelled anti-BSSL antibodies (AS20-AF647) and analyzed by flow cytometry. The presence of BSSL was estimated by mean fluorecence intensity (MFI) of AS20-AF647 staining. (**A**) non-permeablilized (surface staining) and (**B**) permeabilized cells. Red depicts T lymphocytes, blue monocytes and yellow granulocytes. (**C**) Relative MFI values calculated from the histograms in A and B.

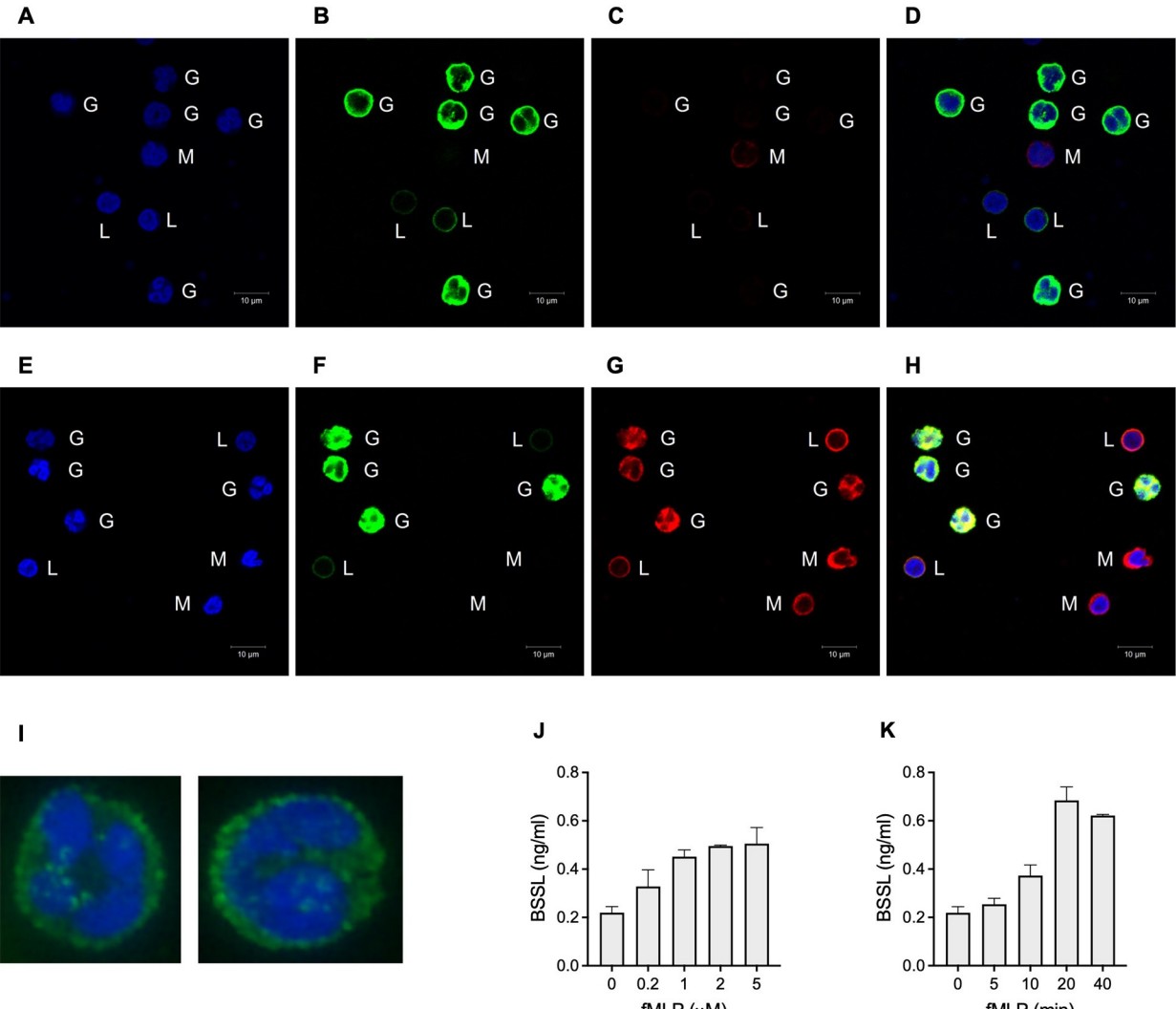

**Fig 2. Immunofluorescent staining of human peripheral blood cells.** To distinguish between cell surface and intracellular localization, cells were either (**A-D**) non-permeabilized or (**E-H**) permeabilized prior to antibody staining. (**A, E**) Cell nuclei stained with DAPI (blue). (**B, F**) Granulocytes (G) and T lymphocytes (L) recognized by anti-CD15 and anti-CD3 antibodies, respectively (both green). Monocytes (M) were recognized by their kidney shaped nucleus and negative CD15 and CD3 staining. (**C, G**) BSSL detected by the polyclonal (monospecific) anti-BSSL antibody Unti (red). (**D, H**) Merged confocal fluorescence images, yellow color indicates co-localization. (**I**) Blow-up showing immunofluorescent staining of human granulocytes using a FITC-labelled anti-BSSL antibody (mAb AS20, green) and confocal microscopy. Cell nucleus stained with DAPI (blue). (**J**) BSSL in culture media secreted from human granulocytes incubated at 37°C for 30 min in the presence of fMLP (0.2–5,0 μM). (**K**) BSSL secreted from granulocytes incubated in the presence of fMLP (1 μM) for indicated time points. Representative data are shown from one of two comparable experiments.

with mAb AS20 at roughly 1:0.5, 1:1 or 1:2 molar ratio before the BSSL/antibody mixture was added to buffy coat, calculated on the bases that the molecular mass of human BSSL (76 kDa) is approximately half of the IgG molecule (150 kDa). The mAb AS20 inhibited (displaced) approximately 80% of bio-BSSL from binding to monocytes (Fig 3A). An irrelevant isotype control antibody (mouse anti-a-synuclein IgG1) did not affect binding of BSSL to leukocytes significantly (Fig 3B).

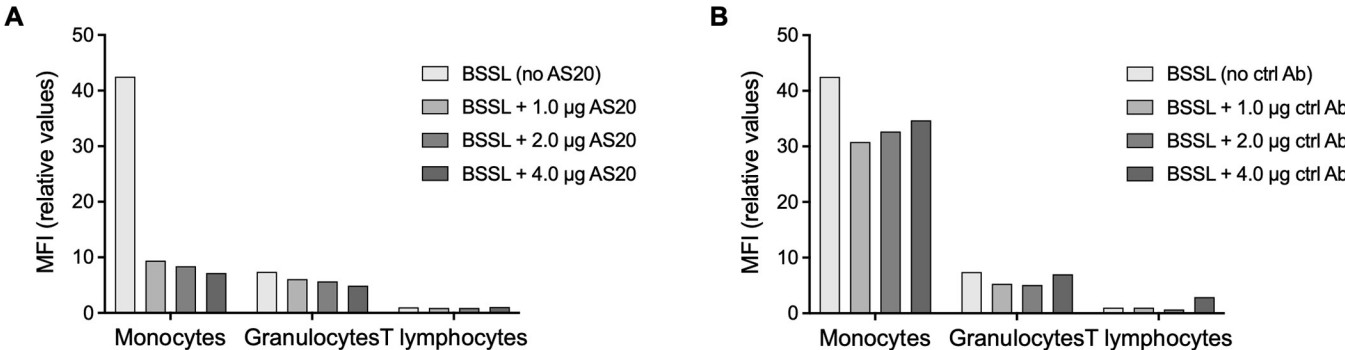

**Fig 3. Binding of BSSL to human leukocytes and displacement with AS20 anti-BSSL antibodies.** Biotin labelled BSSL (1 μg) was incubated with 100 μl of fresh human buffy coat. Binding of bio-BSSL to different cell populations was detected by fluorescently labelled streptavidin in flow cytometery and quantified as median fluorescence intensity (MFI). Monocytes, granulocytes, and T lymphocytes were recognized and gated by fluorescently labelled anti-CD14, anti-CD15 and anti-CD3 antibodies, respectively. The capacity of (**A**) AS20 (1 μg, 2 μg or 4 μg) or (**B**) an isotype control antibody (same concentrations) to displace binding of 1 μg bio-BSSL to monocytes and granulocytes is illustrated as a reduction in MFI compared to the control (no AS20 or isotype control antibody added).

## In vitro cell migration

It was hypothesized that BSSL by interacting with CD14$^+$ monocytes might induce signaling pathways that stimulate cell migration. This was evaluated using an *in vitro* transwell migration assay with human peripheral blood mononuclear cells (PBMCs), freshly prepared from healthy donors. When isolated human milk BSSL was added to the lower compartment at concentrations ranging from 10–300 μg/mL (i.e. approx. 0.1, 0.4, 1.0 and 4.0 μM BSSL) it dose-dependently stimulated migration of PBMCs, particularly CD14$^+$ monocytes, over the semipermeable membrane in the transwell chamber (Fig 4A and 4B). Bovine serum albumin (BSA) at similar protein concentrations, had no effect on total PBMC or CD14$^+$ monocyte migration (Fig 4D and 4E). Neither BSSL, nor BSA showed any effect on migration of CD3$^+$ lymphocytes in the transwell assay (Fig 4C and 4D).

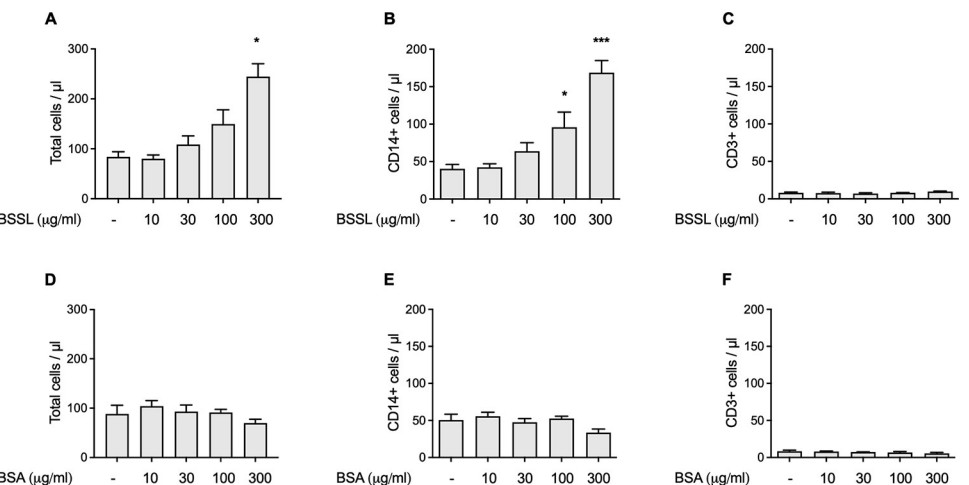

**Fig 4. In vitro cell migration (transwell) assay.** (**A, B**) BSSL stimulates migration of human leukocytes, particularly CD14+ monocytes in a dose-dependent manner. (**D, E**) BSA at similar protein concentrations has no effect on leukocyte migration. (**C, F**) Neither BSSL, nor BSA stimulates migration of CD3+ lymphocytes. Bars show mean ± SEM. *p<0.05; ***p<0.001 compared to baseline (no test item added).

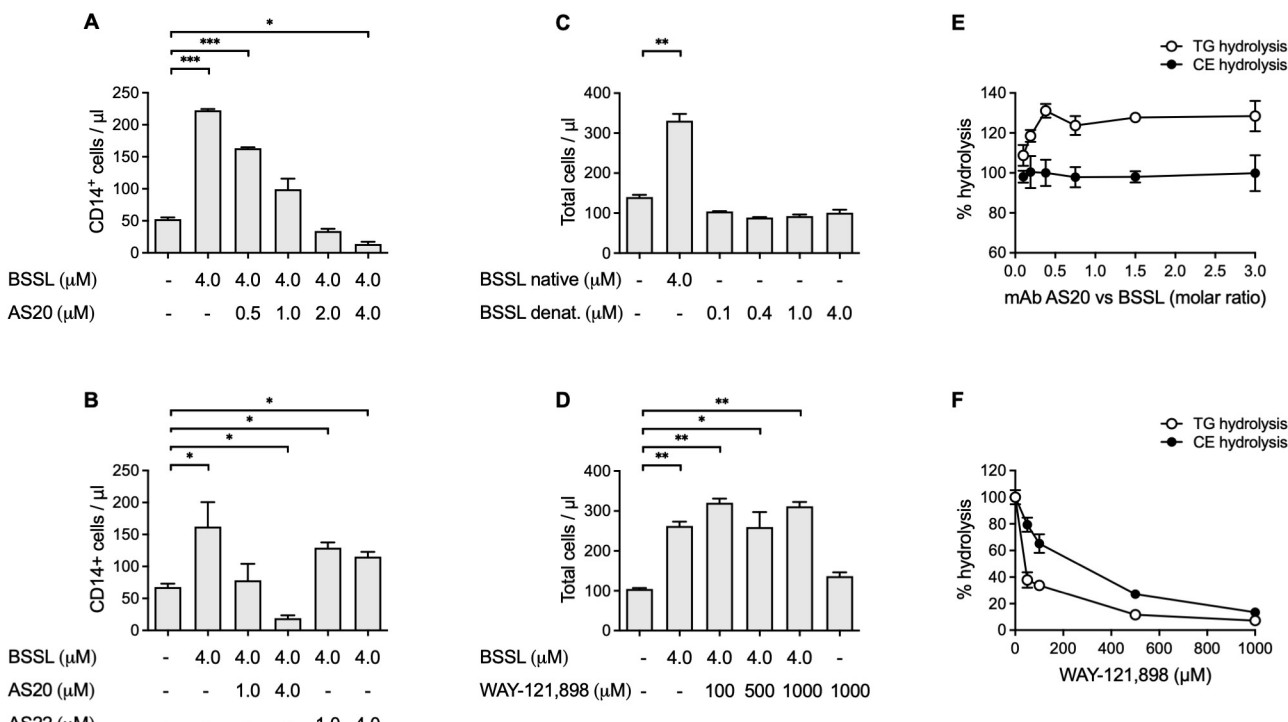

**Fig 5. In vitro cell migration (transwell) assay.** (**A**) BSSL stimulates migration of human CD14+ monocytes and mAb AS20 blocks the stimulatory effect in a dose-dependent manner. (**B**) MAb AS20 but not an isotype control antibody (AS22), blocks BSSL induced cell migration. (**C**) Heat denatured BSSL does not stimulate leukocyte migration. (**D**) The BSSL lipase inhibitor, WAY121,898, does not affect BSSL induced cell migration. (**E**) mAb AS20 does not affect BSSLs capacity to hydrolyze triglycerides and cholesterol esters, whereas (**F**) WAY-121,898 significantly inhibit the enzymatic activity. Bars show mean ± SEM. *p<0.05; **p<0.01; ***p<0.001 compared to baseline (no test item added).

The stimulatory effect of BSSL at 4.0 μM (300 μg/mL) on CD14+ monocyte migration was dose-dependently blocked by mAb AS20. Preincubating BSSL with mAb AS20 at concentrations ranging from 0.5–4.0 μM (75–600 μg/mL) before the BSSL/AS20 antibody mixture was added to the lower compartment of the migration chamber significantly decreased cell migration (Fig 5A). An isotype control antibody (mAb AS22), which binds to BSSL with significantly lower affinity compared to AS20, did not affect BSSL induced cell migration (Fig 5B). Heat denaturation of the BSSL protein (95°C for 10 minutes) totally prevented the stimulatory effect on cell migration (Fig 5C), whereas preincubating BSSL with the carbamate molecule WAY-121,898, an inhibitor of BSSLs enzymatic activity [25], had no additional effect on cell migration even at 250-fold molar excess (Fig 5D).

The effect of mAb AS20 and WAY-121,898 on BSSLs capacity to hydrolyze triglycerides and cholesterol esters was evaluated *in vitro*. The mAb AS20 at concentration ranging from 0.06–1.8 μM had no inhibitory effect on the enzymatic activity of BSSL (45 μg/ml; approx. 0.6 μM), even with mAb AS20 at a 3-fold molar excess (Fig 5E). In contrast, WAY-121,898 at concentrations ranging from 50–1000 μM dose-dependently inhibited the capacity of BSSL (100 μg/ml; approx. 1.3 μM) to hydrolyze triglycerides and cholesterol esters (Fig 5F).

## BSSL in patients with rheumatoid arthritis

The concentration of BSSL was determined in plasma samples from 15 RA patients before and during treatment with the TNF-inhibitor infliximab (Remicade®). The BSSL levels correlated significantly with disease activity score (DAS28), ($r_s$ = 0.47, $p$ = 0.001) (Fig 6A). There was a

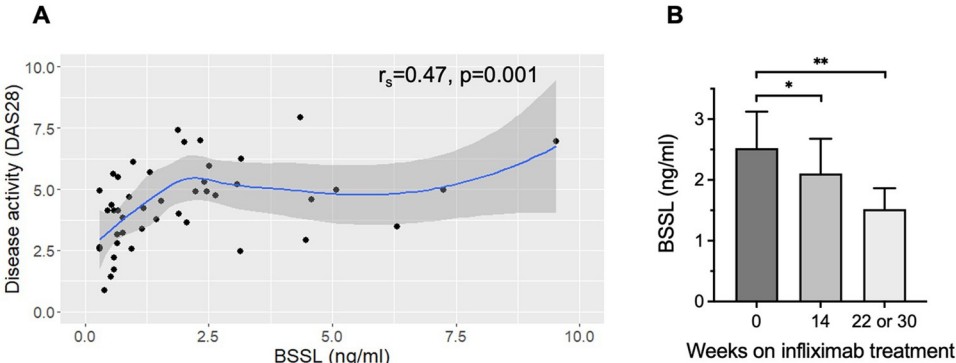

**Fig 6. BSSL plasma levels and disease activity score (DAS28) in 15 RA patients undergoing treatment with infliximab infusions.** BSSL levels and DAS28 were determined at baseline (prior to first infliximab dose) and after 14 and 22 or 30 weeks of treatment. (**A**) The BSSL levels correlate with DAS28. A local regression (LOESS) curve is shown with 95% confidence interval shaded in gray. (**B**) BSSL plasma levels decrease with time during treatment with infliximab. Bars show mean ± SEM. *p<0.05; **p<0.01 compared to baseline.

significant reduction in BSSL plasma levels with duration of infliximab treatment, from 2.5 ± 0.6 ng/mL (mean ± SEM) at baseline (day 1) to 2.1 ± 0.6 ng/mL at 14 weeks (p = 0.05) and 1.5 ± 0.3 ng/mL at 22 or 30 weeks (p = 0.003 compared with baseline values) (Fig 6B). In parallel, there was a decrease in DAS28 over time from 5.7 ± 0.3 at baseline to 3.7 ± 0.4 at 14 weeks (p<0.001) and 3.7 ± 0.3 at 22 or 30 weeks (p<0.001).

The BSSL plasma levels in RA patients also correlated with total leukocyte ($r_s$ = 0.58, $p$ = 0.004) and neutrophil counts ($r_s$ = 0.61, $p$ = 0.002) (Fig 7), and almost, but not significantly with ESR ($r_s$ = 0.38, $p$ = 0.074) and CRP ($r_s$ = 0.40, $p$ = 0.063). There was a strong pairwise correlation between levels of IL-1β, IL-6 and TNF in the RA plasma samples, but the cytokine levels did not significantly change with duration of infliximab treatment and did not correlate with plasma levels of BSSL (S1 Fig).

## BSSL in patients with psoriatic arthritis

The concentration of BSSL determined in serum samples from the 43 PsA patients and 28 healthy controls showed significantly increased BSSL levels from 1.7 ± 0.2 ng/mL in healthy controls, to 2.0 ± 0.2 ng/mL in PsA patients with oligoarthritis ($p$ = 0.018), and to 2.2 ± 0.2 ng/

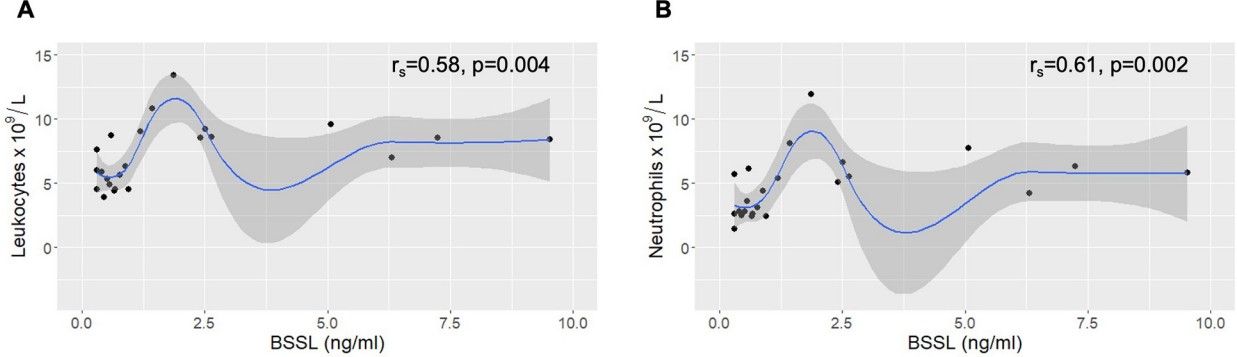

**Fig 7. BSSL levels and leukocyte numbers in plasma from RA patients.** Plasma samples taken from 8 patients at three clinical visits each, i.e. at baseline (before the first infliximab infusion) and after 14 and 22 or 30 weeks of treatment. The BSSL concentration correlates significantly to (**A**) total leukocyte counts and (**B**) neutrophil counts. Local regression (LOESS) curves are shown with 95% confidence intervals shaded in gray.

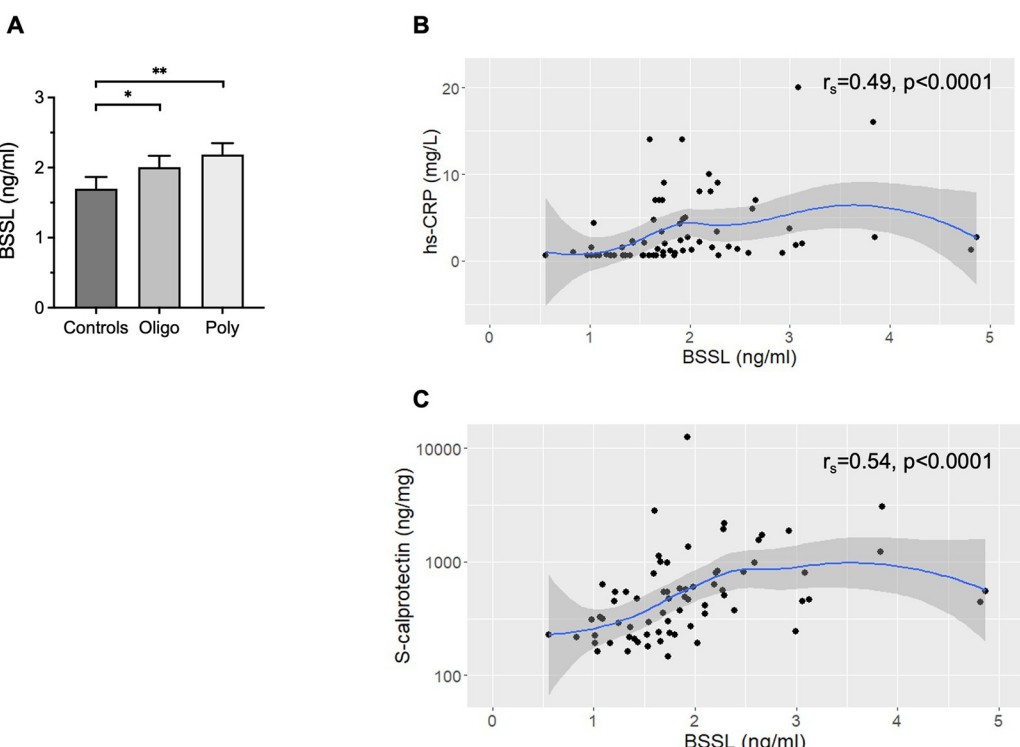

**Fig 8. BSSL serum levels in psoriatic arthritis patients and healthy controls.** (**A**) The concentration of BSSL was significantly higher in serum from PsA patients with both oligoarthritis (n = 22) and polyarthritis (n = 21) compared to healthy controls (n = 28). (**B, C**) The BSSL serum levels correlated significantly to hs-CRP and s-calprotectin. Local regression (LOESS) curves are shown with 95% confidence intervals shaded in gray. Bars show mean ± SEM. *p<0.05; **p<0.01.

mL in patients with polyarthritis ($p = 0.005$) (Fig 8A). The BSSL levels correlated with serum levels of hs-CRP ($r_s = 0.49$, $p<0.0001$) (Fig 8B) and calprotectin (S100A8/A9) ($r_s = 0.54$, $p<0.0001$) (Fig 8C). There was no correlation between levels of BSSL and any of the 11 cytokines or chemokines measured (IL-12, IL-15, IL-16, IL-17A, IL-18, IL-22, IL-23, IL-33, CCL20, CXCL10 and CXCL12) (S3 Table).

Within the group of PsA patients, a correlation between BSSL concentration and hemoglobin (Hb) was found ($r_s = 0.31$, $p = 0.040$). Additional variables and inflammatory markers that were recorded in PsA patients, i.e., leukocyte- or thrombocyte counts, ESR, s-urate, s-creatinine, years of disease duration and years of duration of skin affection did not correlate with BSSL serum levels (S3 Table).

## Discussion

In the present study we explore the presence and function of BSSL in inflammatory cells and demonstrate that BSSL is secreted by neutrophils, binds to monocytes and stimulates their migration *in vitro*. This effect on cell migration seems to be mediated via BSSLs structural properties and physical interaction with monocytes rather than its enzymatic activity. The carbamate molecule WAY-121,898, which is a potent inhibitor of BSSLs enzymatic activity [25], did not interfere with the stimulatory effect of BSSL in the transwell migration assay. In contrast, the mAb AS20, which does not affect BSSLs enzymatic activity but physically blocks BSSL from binding to monocytes, significantly and dose-dependently prevented BSSL induced cell migration in the transwell assay. No specific receptor involved in BSSL-induced signaling

and cell migration has so far been identified. However, BSSL has previously been reported to interact with DC-SIGN, CXCR4, LOX-1 and mannose receptor CD206 [4, 12, 29, 30]. Further studies will show whether any of these receptors are involved in the currently discovered interaction between BSSL and monocytes.

The BSSL protein has previously been recognized as an important player in the inflammatory process of arthritis in rodents [15]. The data presented here clearly demonstrates that BSSL is also closely associated with inflammatory joint diseases in humans. In RA patients, BSSL levels in peripheral blood correlated significantly with disease activity scores and decreased during treatment with a TNF-inhibitor in parallel with decreasing disease activity scores. In PsA patients, BSSL serum levels were significantly higher, particularly in polyarthritis, compared with healthy controls. From these data alone, it is not possible to determine whether BSSL actively stimulates the inflammatory cascade or is merely an inactive biomarker of inflammation. However, previous studies showing that BSSL knockout mice are almost completely protected from developing collagen induced arthritis, and that anti-BSSL antibodies can be used to mitigate disease severity in mice and rat arthritis models [15] clearly supports the first alternative and thus point out BSSL as an exciting novel potential target for new anti-inflammatory drugs. Current therapies such as biologic disease-modifying anti-rheumatic drugs (bDMARDs) have contributed significantly to the improvement of reducing the disease activity and thus, to patients´ quality of life, but they do have adverse effects [31, 32]. Moreover, about one third of RA patients do not respond and 1/3 respond transiently, to these novel biological drugs [33, 34] which further demonstrates the need for new complementary drugs with alternative targets.

The results reported herein have to be considered in the light of some limitations, mainly related to mode of action. Firstly, there is an uncertainty about the origin of circulating BSSL and the local concentrations at a site of inflammation. Previous studies have reported contradictory data whether endogenous BSSL secreted from the exocrine pancreas or orally administered is absorbed from the intestine or not [7, 13, 14]. Hence, although the present study shows that BSSL is secreted from inflammatory cells, it cannot be excluded that circulating BSSL, at least in part, originate from the pancreas. It is however tempting to speculate that BSSL accumulates at sites of inflammation via granulocytes (neutrophils), which are continuously recruited to the inflammatory sites and release various neutrophil granule proteins, including BSSL. A second limitation concerns the study material. All experiments that address the interaction of BSSL with peripheral white blood cells *ex vivo*, and the transwell migration assays were performed with blood and cells from healthy donors. In future studies, the presence and function of BSSL in white blood cells and tissues from healthy donors should be compared with cells and tissues from patients with immune-mediated inflammatory diseases. Assuming that BSSL´s role in inflammation is not limited to diseases with joint inflammation it would also be valuable to repeat measurements of levels of BSSL in serum or plasma in other large cohorts of well-characterized patients with other chronic inflammatory diseases and matched healthy controls as well as BSSL´s presence at site of inflammation.

In summary, the present study confirms that circulating BSSL levels is closely associated with disease activity in inflammatory joint diseases, not only in preclinical animal models but also in human patients with RA and PsA. Although, the mode of action is not yet fully clarified, data presented here propose that BSSL is a proinflammatory component in the innate immune system involved in the recruitment of inflammatory cells to a site of inflammation. When the inflammation persists BSSL becomes a target for treatment of the inflammation. Antibodies towards BSSL prevent excessive cell migration and thereby mitigates inflammation. Thus, the BSSL protein becomes a possible novel target for treatment of chronic inflammation, e.g. RA as previously suggested by animal models. A first in man study of a monoclonal antibody

targeting BSSL was recently initiated (https://clinicaltrials.gov/ct2/show/NCT05576012?term=lipum&draw=2&rank=1).

## Supporting information

**S1 Table. Demographic data of the included patients with RA.**
(DOCX)

**S2 Table. Demographic data of the included patients with PsA.**
(DOCX)

**S3 Table. Correlations between serum BSSL and disease associated parameters in PsA patients and healthy controls.**
(DOCX)

**S1 Fig. Proinflammatory cytokines before and during treatment with infliximab infusions.** (**A-C**) Pariwise correlation between IL-1β, IL-6 and TNFα plasma concentrations measured in 8 RA patients at three clinical visits each, i.e. in total 24 samples taken before the first infliximab infusion and then after 14 and 22 or 30 weeks of treatment. (**D-F**) IL-1β, IL-6 and TNFα levels in plasma samples taken from 8 patients at each of three visits did not change significantly with duration of treament. Bars show mean ± SEM. (**G-I**) There was no correlation between BSSL plasma levels and any of the three cytokines analyzed.
(TIFF)

## Acknowledgments

We would like to acknowledge the staff and the patients participating in the study at the Department of Rheumatology at the University Hospital, Umeå, as well as the staff at the Pediatric research laboratory, Department of Clinical Sciences, Umeå University. We are grateful to Ass. Prof. Torbjörn Lind and B.Sc. Björn Tavelin, Umeå University, for fruitful discussions and assistance with statistical issues.

## Author Contributions

**Conceptualization:** Susanne Lindquist, Gerd-Marie Alenius, Solbritt Rantapää-Dahlqvist, Lennart Lundberg, Olle Hernell.

**Formal analysis:** Susanne Lindquist, Yuhang Wang.

**Funding acquisition:** Susanne Lindquist, Gerd-Marie Alenius, Solbritt Rantapää-Dahlqvist, Olle Hernell.

**Investigation:** Susanne Lindquist, Yuhang Wang, Eva-Lotta Andersson, Shizuko Tsuji Grebe.

**Methodology:** Yuhang Wang, Eva-Lotta Andersson, Shizuko Tsuji Grebe.

**Project administration:** Susanne Lindquist, Olle Hernell.

**Resources:** Susanne Lindquist, Gerd-Marie Alenius, Solbritt Rantapää-Dahlqvist, Olle Hernell.

**Supervision:** Susanne Lindquist, Olle Hernell.

**Visualization:** Susanne Lindquist, Eva-Lotta Andersson.

**Writing – original draft:** Susanne Lindquist.

**Writing – review & editing:** Susanne Lindquist, Yuhang Wang, Eva-Lotta Andersson, Shizuko Tsuji Grebe, Gerd-Marie Alenius, Solbritt Rantapää-Dahlqvist, Lennart Lundberg, Olle Hernell.

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
