## [Decision Letter · Decision Letter 0]

29 May 2023

PONE-D-23-09819Effects of bile salt-stimulated lipase on blood cells and associations with disease activity in human inflammatory joint disordersPLOS ONE

Dear Dr. Lindquist,

Thank you for submitting your manuscript to PLOS ONE. After careful consideration, we feel that it has merit but does not fully meet PLOS ONE’s publication criteria as it currently stands. Therefore, we invite you to submit a revised version of the manuscript that addresses the points raised during the review process.

We look forward to receiving your revised manuscript.

Kind regards,

Masanori A. Murayama

Academic Editor

PLOS ONE

Additional Editor Comments:

Thank you for submitting your study. Reviewers have required revision at many points, especially in various figures. So I will decide "Major Revision" in this time. Please revise.

Reviewers' comments:

Reviewer's Responses to Questions

**Comments to the Author**

1. Is the manuscript technically sound, and do the data support the conclusions?

Reviewer #1: Partly

Reviewer #2: Partly

2. Has the statistical analysis been performed appropriately and rigorously? 

Reviewer #1: Yes

Reviewer #2: I Don't Know

3. Have the authors made all data underlying the findings in their manuscript fully available?

Reviewer #1: No

Reviewer #2: Yes

4. Is the manuscript presented in an intelligible fashion and written in standard English?

Reviewer #1: Yes

Reviewer #2: Yes

5. Review Comments to the Author

Reviewer #1: - figure 8 is lacking in the manuscript;

- in Figure 6A and Figure 7A and B, the regression curve need to be drawn, with confidence interval

- in Figure 6B/ and Figure 5 A to D, the group that are compared for statistical analyses need to be better evidenced

- Figure 6C simply shows that DAS28 is reduced in treated patients...no need to put this part in the manuscrit. eventually is the suppl materials

- Figure 5D; the absence of effect of the WAY-121 inhibitor needs to be explained; While the KO as an effect in animal models, can a KI model, depleting enzymatic activity, confirm this chemical approach ?

- Figure 3A. The absence of effect on granulocytes must be explained

- Figure 2A; confocal images must be shown to prove intracellular staining

Reviewer #2: The Manuscript by Lindquist et al., "Effects of bile salt-stimulated lipase on blood cells and associations with disease activity in human inflammatory joint disorders" tried to shows the correlation between BSSL and arthritic diseases (RA and pSA). The study is preliminary to conclude the hypothesis and need further clarification and submission of missing data (figure 8).

1. Did author looked at the cytokines or chemokines after stimulations of CD14 cells with BSSL.

2. Is there any reason to use different dose and values in fig 4 and 5 for BSSL? Keep the values consistent and also these 2 figures can be merge into one.

3. Is there any value range for BSSL in healthy vs RA or pSA?

4. Fig 8 is missing from the manuscript.

5. Reference 15 was used bylarge in whole manuscript.

6. PLOS authors have the option to publish the peer review history of their article (what does this mean?). If published, this will include your full peer review and any attached files.

Reviewer #1: No

Reviewer #2: No

---

## [Author Response · Author response to Decision Letter 0]

9 Jul 2023

Response to Editor (Journal requirements):

- Please ensure that your manuscript meets PLoS ONE’s style requirements.

Response: We have corrected some discrepancies, including file names.

- We note that you have included the phrase “data not shown” in your manuscript. 

Response: The data referred to has been included in Fig 5 (E-F) in this revised manuscript.

Reviewers’ Comments to Author:

Reviewer #1:

- Figure 8 is lacking in the manuscript.

Response: We apologize for not having noticed that Figure 8 was missing in the original submission. This figure is now included.

- In Figure 6A and Figure 7A and B, the regression curves need to be drawn, with confidence interval.

Response: The figures have been updated with nonparametric local regression curves (LOESS) and 95% confidence intervals.

- In Figure 6B/ and Figure 5 A to D, the group that are compared for statistical analyses need to be better evidenced.

Response: Connecting lines have been drawn to indicate which bars are compared.

- Figure 6C simply shows that DAS28 is reduced in treated patients...no need to put this part in the manuscript. eventually is the suppl materials.

Response: We agree, the figure 6C has been withdrawn.

- Figure 5D; the absence of effect of the WAY-121 inhibitor needs to be explained; While the KO as an effect in animal models, can a KI model, depleting enzymatic activity, confirm this chemical approach?

Response: The text in the discussion (first paragraph) has been slightly modified to try to clarify, and data showing the effect of WAY-121,898 on BSSLs enzymatic activity has been added in Fig 5 (E-F). The reviewer’s suggestion to develop a KI model, depleting enzymatic activity, is a great idea and certainly something that we will consider in our further research planning. 

- Figure 3A. The absence of effect on granulocytes must be explained.

Response: Exogenous BSSL binds primarily to the surface of monocytes and only a weak binding was detected to granulocytes. Hence, although a minor (dose dependent) effect of mAb AS20 can be seen on the binding to granulocytes, we cannot expect an effect as clear as for monocytes. 

- Figure 2A; confocal images must be shown to prove intracellular staining.

Response: Additional data with immunofluorescent staining’s and confocal images have been added to the manuscript (Fig 2). A new paragraph entitled “Immunofluorescence staining and confocal imaging” has been included in Material and Methods.

Reviewer #2:

The Manuscript by Lindquist et al., "Effects of bile salt-stimulated lipase on blood cells and associations with disease activity in human inflammatory joint disorders" tried to show the correlation between BSSL and arthritic diseases (RA and PsA). The study is preliminary to conclude the hypothesis and need further clarification and submission of missing data (figure 8).

- Did author look at the cytokines or chemokines after stimulations of CD14 cells with BSSL.

Response: This is certainly an interesting question that we currently address as part of a future manuscript. We ask for the reviewer´s understanding in this regard.

- Is there any reason to use different dose and values in fig 4 and 5 for BSSL? Keep the values consistent and also these 2 figures can be merged into one.

Response: In Fig 4 we use BSA as a control to show that the effect on cell migration is specific to BSSL, and not an unspecific effect of any unrelated protein at similar concentrations. In Fig 5 we investigate the capacity of different test items (antibodies and WAY-121,898) to interfere with BSSL induced cell migration. In these experiments the molar ratios between BSSL and test items are most relevant, and therefore we report concentrations in µM instead of µg/ml. A few changes have been made in the last paragraph on p 15 to clarify the relationship between protein concentration (µg/ml) and molarity, i.e., BSSL at 300 µg/ml = 4 µM.

- Is there any value range for BSSL in healthy vs RA or PsA?

Response: These data are shown in Fig 6 (RA) and Fig 8 (PsA and healthy). Fig 8 was unfortunately missing in the original submission but is now included.

- Fig 8 is missing from the manuscript.

Response: Again, we apologize for not having noticed that Figure 8 was missing in the original submission. This figure is now included.

- Reference 15 was used by large in whole manuscript.

Response: This point is well taken. However, to the best of our knowledge there are no other papers in the literature that describe the role of BSSL in animal models of inflammatory arthritis.

---

## [Decision Letter · Decision Letter 1]

31 Jul 2023

Effects of bile salt-stimulated lipase on blood cells and associations with disease activity in human inflammatory joint disorders

PONE-D-23-09819R1

Dear Dr. Susanne Lindquist,

We’re pleased to inform you that your manuscript has been judged scientifically suitable for publication and will be formally accepted for publication once it meets all outstanding technical requirements.

Kind regards,

Masanori A. Murayama

Academic Editor

PLOS ONE

Additional Editor Comments (optional):

Thank you for submitting revised manuscript. All reviewers recommend the decision "Accept". Congratulations.

Reviewers' comments:

Reviewer's Responses to Questions

**Comments to the Author**

1. If the authors have adequately addressed your comments raised in a previous round of review and you feel that this manuscript is now acceptable for publication, you may indicate that here to bypass the “Comments to the Author” section, enter your conflict of interest statement in the “Confidential to Editor” section, and submit your "Accept" recommendation.

Reviewer #1: All comments have been addressed

Reviewer #2: All comments have been addressed

2. Is the manuscript technically sound, and do the data support the conclusions?

Reviewer #1: Yes

Reviewer #2: Yes

3. Has the statistical analysis been performed appropriately and rigorously? 

Reviewer #1: I Don't Know

Reviewer #2: Yes

4. Have the authors made all data underlying the findings in their manuscript fully available?

Reviewer #1: Yes

Reviewer #2: Yes

5. Is the manuscript presented in an intelligible fashion and written in standard English?

Reviewer #1: Yes

Reviewer #2: Yes

6. Review Comments to the Author

Reviewer #1: I'm not familiar with the tests that were used to analysze correlation data. othewise, all my comments were addressed.

Reviewer #2: As the author responded to the queries raised previously. I have no further question and my recommendation for acceptance.

7. PLOS authors have the option to publish the peer review history of their article (what does this mean?). If published, this will include your full peer review and any attached files.

Reviewer #1: **Yes: **Philippe Georgel

Reviewer #2: No

---

## [Editor Report · Acceptance letter]

4 Aug 2023

PONE-D-23-09819R1 

Effects of bile salt-stimulated lipase on blood cells and associations with disease activity in human inflammatory joint disorders 

Dear Dr. Lindquist:

I'm pleased to inform you that your manuscript has been deemed suitable for publication in PLOS ONE. Congratulations! Your manuscript is now with our production department. 

Kind regards, 

on behalf of

Dr. Masanori A. Murayama 

Academic Editor

PLOS ONE